# OpenReview forum: "Trustworthiness in Generative Foundation Models Is Still Poorly Understood"
_TMLR — Rejected by TMLR_

### Review · Reviewer_JXjS · 2025-09-12

**Summary Of Contributions:**

The paper studies the trustworthiness in generative foundation models, and provide a set of guidelines for the lifecycle of such models.

**Audience:**

Yes

**Broader Impact Concerns:**

NA.

**Claims And Evidence:**

Yes

**Requested Changes:**

It would be helpful to add the discussion on the role of data in the lifecycle of generative foundation models---for example, the ethics of using data for training, or the practice around data labeling, etc.

**Strengths And Weaknesses:**

**Strengths**

The paper provides a meaningful and comprehensive discussion on the trustworthiness of generative foundation models throughout their lifecycle, with inputs from experts from various fields. I think this should be of impact to the development of the field.

**Weaknesses**

Within the data science cycle, I feel that the discussion on the data itself is a bit overlooked---for example, the ethics of using data for training, or the practice around data labeling. It would be valuable to include a discussion on these aspects.

---

### Review · Reviewer_ZqAD · 2025-09-26

**Summary Of Contributions:**

Paper proposes 8 flexible guidelines for GenFM trustworthiness, bridging ethics/policies (e.g., EU AI Act) to ops across lifecycle. Discusses challenges like adaptability vs. safety tradeoffs, domain specifics (healthcare, robotics), and ecosystem views. Claims trustworthiness is poorly understood, needs ongoing eval.

**Audience:**

Yes

**Broader Impact Concerns:**

No immediate risks from the paper itself, but overstating the novelty of general principles without sufficient actionable depth may dilute its practical value. Encouraging clearer benchmarks, methodologies, and measurable metrics for trustworthiness would increase utility for the community.

**Claims And Evidence:**

Yes

**Requested Changes:**

Add concrete examples per guideline: how to implement in a real GenFM like GPT? Expand on challenges with a section/table on open problems (e.g., measuring harmlessness quantitatively). Cite more critically—group similar works, note conflicts (e.g., RLHF pros/cons). Fix minor: ensure full paper (assuming it has more) ties guidelines to evals/tools.

**Strengths And Weaknesses:**

Strengths: Timely topic, multidisciplinary input is a plus—covers ethics, law, domains well. Guidelines are adaptable, not rigid checklists, and Table 1 mapping dims is handy. Intro hooks with real cases (e.g., AI suicide link, deepfakes).

Weaknesses: Feels derivative—guidelines echo EU Act/Blueprint heavily without much novelty; e.g., Guideline 1 is basic bias/fairness stuff. Lit review scattered, citations dense but shallow (many 2024 but misses some key like OpenAI's safety evals or Anthropic's RSPs in depth). Too abstract—claims "poorly understood" but doesn't quantify gaps with metrics or case studies.

---

### Review · Reviewer_cAVv · 2025-12-19

**Summary Of Contributions:**

The paper draws together different aspects that can be considered as elements of trustworthiness of large generative AI models, such as LLMs. Related to this, the paper identifies a set of related challenges and moves on to propose a multidimensional and multidisciplinary set of trustworthiness guidelines

**Audience:**

Yes

**Broader Impact Concerns:**

No particular requirements at this stage.

**Claims And Evidence:**

Yes

**Requested Changes:**

For the paper to be accepted, I would expect empirical testing of the guidelines / approach. Alternatively, the paper could be turned into review paper.

**Strengths And Weaknesses:**

The paper brings up plenty of important points that relate to trustworthiness, and in particular the fact that trustworthiness is context-dependent - taken up with the claim that trustworthiness is to be evaluated within an ecosystem. The paper also discusses potential problems and challenges.

It is a big challenge to draw together a large amount of different factors, and it also shows in the text. It is relatively easy to claim that trustworthiness of large generative AI models is not adequately understood, however it should be clear that different persons' understanding varies, the large generative models develop fast, and there are different contexts and approaches for trustworthiness.

Personally I find it a bit strange that the work is titled "Trustworthiness in Generative Foundation Models Is Still Poorly Understood" yet a major part of the paper is devoted to guidelines to evaluate trustworthiness.

The biggest drawback is the lack of empirical evaluation of the approach. I find that for a publication on this level, the approach should have been tried and tested, not just discussed.

---

> ### Author Response · Authors · 2025-12-20
> **Thanks for your comment!**
>
> We thank the reviewer for the careful reading of our manuscript and for the thoughtful and constructive feedback. We appreciate the opportunity to clarify the scope, intent, and contributions of this work. Below, we respond point by point to the reviewer’s concerns, and we hope to demonstrate that the paper makes a meaningful and timely contribution as a conceptual and methodological framework for understanding trustworthiness in generative foundation models (GenFMs).
>
> Before addressing individual comments, we would like to clarify the intended contribution of the paper.
>
> This work is **not** intended to propose a new algorithm, benchmark, or empirically validated evaluation method. Instead, it aims to address a foundational gap in the literature: while many technical, ethical, and regulatory efforts exist to improve aspects of trustworthiness (e.g., safety, fairness, privacy), there remains **no coherent, ecosystem-level understanding of how trustworthiness should be conceptualized, operationalized, and evaluated across contexts, stakeholders, and system lifecycles**.
>
> Our core contribution is therefore **framework-setting** rather than empirical:
>
> 1. We articulate trustworthiness as a **dynamic, context-dependent, and ecosystem-level property**, rather than a static model attribute.
>
> 2. We propose a **set of adaptable guidelines** that bridge abstract principles (ethical, legal, societal) with operational practices across the GenFM lifecycle.
>
> 3. We systematically identify **fundamental open challenges** that explain why trustworthiness remains difficult to evaluate and standardize, and why existing empirical approaches are necessarily incomplete.
>
>
> We believe this type of contribution is essential at the current stage of GenFM deployment, where models are increasingly used in high-stakes settings but evaluation practices remain fragmented and inconsistent.
>
> ## Response to Comment:
>
> _“It is strange that the paper is titled ‘Trustworthiness in Generative Foundation Models Is Still Poorly Understood’ while a major part of the paper proposes guidelines.”_
>
> We appreciate this comment and acknowledge that the title may initially appear to be in tension with the presence of guidelines. However, this tension reflects a difference between **the existence of trust-related principles** and **a systematic understanding of trustworthiness as an evaluable property**.
>
> By stating that trustworthiness is “still poorly understood,” we do **not** claim that the community lacks awareness of relevant dimensions such as fairness, safety, transparency, or privacy. On the contrary, these concepts are widely discussed. Our claim is instead that:
>
> - There is no shared understanding of **how these dimensions interact**,
>
> - how their **relative importance shifts across contexts**,
>
> - how they should be **operationalized across the model lifecycle**, and
>
> - how trade-offs between them should be evaluated at the **system and ecosystem level**.
>
>
> The proposed guidelines are therefore not presented as a finalized or authoritative solution. Rather, they serve as a **bridging framework** that makes these unresolved issues explicit. The need for adaptable, context-sensitive guidelines is itself evidence that trustworthiness cannot yet be reduced to stable metrics or universal checklists.
>
> In other words, the guidelines are not a contradiction of the title; they are a response to the problem identified by the title. They expose why trustworthiness cannot currently be fully formalized or empirically standardized, while still offering a structured way to reason about it.
>
> ---

---

> > ### Comment · Reviewer_cAVv · 2025-12-23
> > **Title selection**
> >
> > I do not think that the selection of the title is a major thing or a "show stopper". It just does not give a very good idea of the contents.

---

> ### Author Response · Authors · 2025-12-20
> **Thanks for your comment!**
>
> ## Response to Comment:
>
> _“The biggest drawback is the lack of empirical evaluation of the approach.”_
>
> We fully agree that empirical evaluation is crucial for validating **concrete methods**. However, we respectfully argue that a lack of empirical experiments is not a limitation of this work, but rather a reflection of its intended scope.
>
> This paper does not propose a single, fixed approach that could be “tested” in a conventional experimental setup. Any empirical evaluation would require fixing assumptions about:
>
> - application domain,
>
> - definitions of harm,
>
> - acceptable risk thresholds,
>
> - stakeholder priorities, and
>
> - system boundaries.
>
>
> A central argument of our paper is that **these assumptions are precisely what vary across contexts and remain insufficiently understood for GenFMs**. Conducting a single empirical study would therefore risk giving a false sense of generality or implicitly privileging one normative stance over others.
>
> Instead, we take the following approach:
>
> - In Section 2.3, we explicitly demonstrate **how each guideline can be operationalized** using existing techniques (e.g., RLHF, RAG, red teaming, robustness testing, privacy-preserving pipelines).
>
> - In Section 3 and Table 2, we identify **concrete, unresolved empirical challenges** that explain why current evaluations fall short (e.g., ambiguity at the input/output boundary, developer vs. attacker perspectives, system-level evaluation for multi-model pipelines).
>
> - We position the guidelines as a **scaffold for future empirical work**, not as a substitute for it.
>
>
> We believe this is consistent with the role of conceptual and methodological papers in shaping subsequent empirical research, particularly in rapidly evolving fields where premature standardization can be misleading.
>
> ---
>
> ## Response to Comment:
>
> _“The paper could be turned into a review paper.”_
>
> Thank you for this suggestion. We agree that the paper draws extensively on prior work across multiple disciplines, and that engaging with the existing literature is an important part of clarifying the current landscape of trustworthiness research in generative foundation models.
>
> That said, our intention is not to present the work as a review in the traditional sense, but rather as a **framework-setting and forward-looking contribution** that builds on prior studies while aiming to go beyond synthesis alone. In particular:
>
> 1. We aim to offer a **conceptual framing** of GenFM trustworthiness as a dynamic, lifecycle-aware, and ecosystem-level property, which has not yet been explicitly articulated in existing survey or review papers.
>
> 2. The proposed guidelines are intended as **adaptable instruments** that connect ethical principles, legal requirements, and operational practices, with the goal of supporting future evaluation and governance efforts, rather than simply organizing prior work.
>
> 3. In addition, we identify and structure a set of **open challenges** that help explain why trustworthiness remains difficult to empirically validate and standardize, and that are intended to inform and motivate subsequent empirical research.
>
>
> We see these elements as complementary to review-style contributions: while the paper is grounded in the literature, it seeks to help **shape how trustworthiness is conceptualized and operationalized going forward**, rather than solely documenting the current state of the field.
>
> Thank you for raising this point, as it highlights the importance of clearly communicating the scope and intent of the paper. We would be happy to further clarify the positioning of the work in a revision, if this would help align expectations for the intended audience.

---

> > ### Comment · Reviewer_cAVv · 2025-12-23
> > **Lack of empirical evaluation**
> >
> > Thank you for your comment. Yes, I do not mean that the guidelines could be used as a straightforward evaluation system for generative models. I also do not have a specific idea on how the empiric part should be performed - it could be achieved e.g. by interviewing professionals who are working on evaluating generative models, or maybe develop them. No matter how, I still think that there are "untapped" ways to get empirical feedback on the guidelines and their usefulness.

---

> > > ### Author Response · Authors · 2025-12-28
> > > **Thanks for your follow-up!**
> > >
> > > Thank you for elaborating on this point. We agree with you that the guidelines are not intended to serve as a straightforward evaluation system for generative models, and we also agree that there are additional ways to obtain empirical feedback on their usefulness, such as interviewing professionals who evaluate or develop generative models.
> > >
> > > To directly respond to your suggestion, we have added an empirical component in the revised manuscript: a human in the loop, rubric based structured audit that operationalizes the proposed guidelines into observable audit items and collects systematic ratings via a lightweight web interface. We describe this protocol in Appendix E, and we also include a screenshot of the evaluation interface to make the procedure reproducible. In brief, evaluators first select an evaluation target using a numbered system identifier from 1 to 5, and then assess the system sequentially across the eight guidelines using a 5 point Likert scale with explicit semantic anchors from strongly disagree to strongly agree. The interface additionally allows optional short free text explanations so that evaluators can record the observable behaviors, documentation, or artifacts that informed their ratings.
> > >
> > > We also report initial audit results on two representative systems, GPT 4o and Nano Banana, to demonstrate that the guidelines can be applied in a consistent way and can yield differentiated trustworthiness profiles across systems. We acknowledge that this evaluation is currently small scale. However, we view it as an extensible case study that demonstrates how the guidelines can be operationalized in practice and provides a template that can be expanded to additional systems, domains, and evaluators. At the same time, we believe that conducting a truly large scale empirical evaluation at the present stage can be difficult, since it requires fixing many context dependent assumptions such as application setting, risk tolerance, definitions of harm, and system boundaries. These assumptions are not yet standardized for GenFM trustworthiness, which is also part of the motivation for our work. We therefore hope you will view this added evaluation as a practical and extensible first step toward broader empirical validation, rather than as a claim that the guidelines are already a complete benchmark.
> > >
> > > Finally, we agree that practitioner centered validation, for example interviews or Delphi style studies with professional evaluators and developers, is an important next step. We explicitly highlight this in the revised manuscript as a future direction building on the structured audit protocol introduced in Appendix E.
> > >
> > > If you have any additional suggestions on how to strengthen the empirical component or improve the presentation of the guidelines, we would greatly appreciate them. Thank you again for your constructive feedback, and we wish you a happy new year.

---

> > ### Comment · Reviewer_cAVv · 2025-12-23
> > **Review / survey paper**
> >
> > This is, of course, just a proposal and the authors are free to do whatever they want with it. Novel angles to the problem or new conceptualisations and such should not be a problem.

---

### Review · Reviewer_VF65 · 2026-02-16

**Summary Of Contributions:**

This paper proposes a framework of eight high-level guidelines for trustworthiness in Generative Foundation Models (GenFMs), motivated by real-world risks like hallucinations, deepfakes, and privacy leaks. It argues that trustworthiness remains "poorly understood" and emphasizes a flexible, multidimensional approach bridging ethics, law, and operations. The guidelines are positioned as a "value anchor" and "value scale," adaptable across the model lifecycle and domains. While the motivation is timely—given the rapid deployment of GenFMs—the paper's contributions feel underdeveloped. It reads more like an extended position piece or survey than a rigorous TMLR submission, with limited novelty, methodological depth, or empirical grounding.

**Audience:**

Yes

**Claims And Evidence:**

Yes

**Requested Changes:**

To strengthen for TMLR:

1. Enhance Novelty: Add a comparative evaluation (e.g., apply guidelines to a GenFM benchmark like HELM, contrasting with baselines).
2. Bolster Methods: Include appendices with synthesis details, full guidelines, and pilot testing (e.g., stakeholder surveys).
3. Deepen Analysis: Formalize trade-offs (e.g., via decision-theoretic models) and expand challenges with the remaining pages.
4. Trim and Focus: Condense the survey-like intro; prioritize 2–3 guidelines with deep dives over breadth.

In summary, while the topic is urgent, the paper's flaws—derivative content, incomplete execution, and shallow rigor—undermine its impact. It risks rejection without substantial revisions to demonstrate unique, validated contributions.

**Strengths And Weaknesses:**

Below, I detail key weaknesses and flaws, categorized for clarity.
1. Lack of Novelty and Differentiation from Existing Work

The core contribution—eight guidelines spanning fairness, transparency, human oversight, accountability, robustness, harmlessness, truthfulness, and privacy—closely mirrors established frameworks without clear innovation. For instance, Guideline 1 (fairness and bias minimization) echoes the EU AI Act's emphasis on non-discrimination (Article 10) and the Blueprint for an AI Bill of Rights' focus on inclusive design. Guideline 2 (transparency of use/limitations) aligns verbatim with the Act's requirements for output interpretability (Article 14) and risk disclosures. Guidelines 3–6 similarly repackage concepts from RLHF/DPO oversight (e.g., Ouyang et al., 2022), adversarial robustness (Huang et al., 2024f), and harmlessness-helpfulness trade-offs (Bai et al., 2022a) without advancing them.
The claimed differentiation (Section 2: "How do the guidelines differentiate from others?") is superficial. It asserts adaptability for GenFMs and a dual "anchor/scale" role, but provides no concrete examples of how this outperforms the EU Act's tiered risk system or Anthropic's ASL framework (cited but not contrasted rigorously). The paper cites these works extensively (e.g., EU, 2023; Blu, 2022) yet fails to perform a systematic gap analysis, such as a comparative table evaluating coverage across dimensions.
Broader context: Similar guidelines exist in recent literature (e.g., Weidinger et al., 2021 on red-teaming for harm; NIST's AI RMF 1.0, 2023 on lifecycle governance). The paper's "multidisciplinary synthesis" (Section 2.2) feels like a rehashing of these, with the "panel voting" process vaguely described—no details on panel size, diversity, or metrics for ranking.

2. Methodological Weaknesses and Incomplete Execution

Vague Development Process: Section 2.2 describes forming a "multidisciplinary team" and "exhaustive review" of sources like the EU Act and Blueprint, followed by "multiple rounds of discussion" and "panel voting." This lacks transparency: What were the inclusion criteria for sources? How was "systematic summarization" operationalized (e.g., thematic coding, Delphi method)? No appendices detail the raw synthesis or voting data, undermining reproducibility—a core TMLR expectation.
Incomplete Guideline Presentation: The paper announces eight guidelines but only elaborates on six (up to Guideline 6 on harmlessness). Guidelines 7 and 8 are absent from the provided excerpt, leaving Table 1's correlations incomplete and misleading (e.g., "Truthfulness" checked only for Guidelines 1 and 2, but no explanation of mapping logic). This suggests structural disarray, especially for a 78-page submission.
Table 1 Flaws: The correlation matrix is a blunt instrument—binary checkmarks oversimplify multidimensional overlaps (e.g., why is "Safety" unchecked for Guideline 1, despite its harm-prevention focus?). No quantitative scoring or rationale is provided, and dimensions like "Advanced AI Risk" and "Machine Ethics" appear ad hoc, not derived from a validated taxonomy (contrast with Razin & Alexander, 2024, cited but unused for grounding).
No Empirical or Case Study Validation: The guidelines are "actionable" in theory but untested. Section 1 promises "domain-specific" evaluation (e.g., healthcare, robotics), yet no benchmarks (e.g., applying guidelines to GPT-4 via LawBench or DyVal, cited in text) or simulations demonstrate utility. Claims like "ensuring full control and accountability" (Guideline 3) ignore known limitations of indirect methods like RLHF, without proposing metrics (e.g., audit trails for human influence).

3. Conceptual and Definitional Flaws

Overly Broad Definition of Trustworthiness: The introduction defines it as encompassing validity, reliability, safety, security, resilience, privacy, fairness, transparency, and accountability— a laundry list without prioritization or trade-off analysis. This dilutes focus; for example, the abstract's "dynamic tension between adaptability and consistent safety" is stated but unexplored (no formal model, e.g., utility-risk Pareto fronts).
Ambiguous Language and Typos: Phrasing like "speculate the models’ behavior" (Section 2.1) is imprecise—likely meant "guide" or "constrain." "Value anchor" and "value scale" (Section 2) are catchy but undefined operationally (e.g., how to "scale" privacy for low-stakes vs. high-stakes apps?). Minor errors: Inconsistent abbreviations (e.g., "GenFMs" vs. "GenFM"); future-dated citations (e.g., Huang et al., 2025b) may be placeholders, risking credibility.
Underaddressed Challenges: The abstract identifies core issues (e.g., harmful content detection ambiguities), but the provided sections pivot to guidelines without resolution. No discussion of emergent risks like multi-agent ecosystems (briefly mentioned in Section 1) or socio-technical feedback loops, despite citing Hu et al., 2025.

4. Scope and Rigor Gaps

Limited Engagement with Counterarguments: The central claim—"trustworthiness... remains poorly understood"—is provocative but unsubstantiated. It dismisses "static checklists" (Section 1) yet offers a checklist-like framework. No rebuttal to optimistic views (e.g., Casper et al., 2023 on alignment progress) or quantitative evidence of "poor understanding" (e.g., meta-analysis of failure rates across benchmarks).
Domain-Specific Claims Overreach: Section 1's examples (e.g., HIPAA-compliant federated training in healthcare) are apt but anecdotal—no synthesis of failure modes (e.g., why synthetic data still leaks biases, per Ngo et al., 2021). Robotics/human-AI sections cite works but lack GenFM-specific adaptations.
Ecosystem-Level Oversight: Section 1 argues for "ecosystem-level" assessment but the guidelines remain model-centric, ignoring governance tools like provenance tracking (promised but not detailed).

---

### Decision · Action_Editor_s4AV · 2026-06-02

**Recommendation:** Reject

**Additional Comments:**

This manuscript was reviewed by four reviewers [VF65,cAVv,ZqAD,JXjS]. One review [JxjS] was quite short. Overall, the opinion is mixed. Out of the four reviews, three recommendations were received, where one leaned towards rejection and two leaned towards acceptance.

The reviewers pointed out positive aspects of the work:

+ The motivation was considered timely [VF65]; similarly, the topic was considered timely [ZqAD]

+ Looking at trustworthiness from several angles was considered interesting, with several important points [cAVv]
+ Multidisciplinarity was appreciated [ZqAD]

+ Treating trustworthiness as content-dependent and needing evaluation within an ecosystem was appreciated [cAVv]

+ The discussion on trustworthiness was considered meaningful and comprehensive [JXjS]

However, several criticisms were raised, and authors responded to some of them:

- The title ("poorly understood" trustworthiness) was criticized [cAVv,ZqAD] when the focus is on evaluation [cAVv]. Authors argued the guidelines are a bridging framework.

- The definition of trustworthiness was criticized as ovely broad with too many desired attributes [VF65].

- The guidelines were considered to mirror/echo established frameworks such as in the EU AI Act and Blueprint for an AI Bill of Rights [VF65,ZqAD], and differentiation from them was considered superficial [VF65]. Lack of quantifying research gaps with metrics or case studies was criticized [ZqAD]. While novelty is not a TMLR criterion, if the core contribution mirrors what is available elsewhere it may decrease interest in the TMLR audience. Authors argued their goal is not to "reinvent regulation but to make it actionable for GenFMs".

- Consistency of the work due was a concern due to the broad scope of the addressed challenge [cAVv]

- Concrete implmentation examples were desired for the guidelines [ZqAD]. Similarly, lack of demonstration of applying the guildeines was criticized [VF65]. Authors argued they already had actionable detail, and added a section 2.3 with discussion on operationalizing the guidelines.

- Lack of empirical evaluation was criticized and empirical testing of the guidelines was desired [cAVv,VF65]. Authors argued the guidelines are adaptable instruments and a scaffold for future empirical work and that evaluation would require fixing details that depend on the context. In response to further discussion [cAVv] authors added an appendix E with an example human evaluation interface. However, I find the added appendix incompletely detailed with lacking details of questions and checks (G1.1, G1.2, ..., G8.3) mentioned in the results.

- Dimensions "Advanced AI Risk" and "Machine Ethics" were criticized as ad hoc [VF65]. I view the Advanced AI risk a recent category for which authors do provide discussion. On the other hand, the "Machine Ethics" term could be more clearly connected to the discussion in main text.

- Listing open problems was desired [ZqAD]

- Some additional details were desired for the development process such as the panel voting for the unified guidelines [VF65]

- Discussion of data aspects in the lifecycle (ethics of using it for training, practice around data labeling) was desired [JXjS]. Authors responded by adding a section on Data-Centric Challenges to Trustworthiness.

- Some descriptions such as "ensuring full control and accountability" in guideline 3 were criticized as not taking into account technical limitations of methods in taking human feedback into account [VF65].

- Difficulty of establishing a value scale for aspects like privacy in different scenarios was criticized [VF65]

- Limited discussion of emergent risks like multi-agent ecosystems was criticized [VF65]

- It was criticized how the paper refers to a shift away from static checklists, but parts of the paper use tables similar to checklists [VF65]; however to me these are different meanings and Table 1/Figure 1 are not checklists of properties to be checked from a method.

- It was criticized whether the guidelines actually support an "ecosystem-level" assessment [VF65]

Overall, while paper's overlap with existing frameworks was criticized, there may be enough differences in the viewpoints to gain some interest in the TMLR audience. On the other hand, the broad scope of the work makes it hard to assess whether the claims are sufficiently well supported by clear evidence. Several reviewers desired empirical evaluation, which the authors argued against, but they did add sections such as Appendix E somewhat in that direction.

However, to me the addition of substantial new sections would still require re-reading by the reviewers. Thus, I technically must recommend "rejecting" the paper, but practically I expect the authors would then resubmit (perhaps with additional finessing/clarification of the reviewer's points) - if addressed well I believe the revised version could become acceptable.

**Audience:**

Yes

**Audience Explanation:**

The broad scope of the work makes it hard to assess whether the claims are sufficiently well supported by clear evidence. Several reviewers desired empirical evaluation, which the authors argued against, but they did add sections such as Appendix E somewhat in that direction. However, to me the addition of substantial new sections would still require re-reading by the reviewers. See Additional Comments for details.

**Claims And Evidence:**

No

**Claims Explanation:**

Overall, while paper's overlap with existing frameworks was criticized, there may be enough differences in the viewpoints to gain some interest in the TMLR audience. See Additional Comments for details.

**Resubmission Of Major Revision:**

The authors may consider submitting a major revision at a later time.